# A Systematic Review of the Clinical Diagnosis of Transient Hypogammaglobulinemia of Infancy

**DOI:** 10.3390/children10081358

**Published:** 2023-08-08

**Authors:** Angel A. Justiz-Vaillant, Trudee Hoyte, Nikao Davis, Candice Deonarinesingh, Amir De Silva, Dylan Dhanpaul, Chloe Dookhoo, Justin Doorpat, Alexei Dopson, Joash Durgapersad, Clovis Palmer, Odalis Asin-Milan, Arlene Faye-Ann Williams-Persad, Rodolfo Arozarena-Fundora

**Affiliations:** 1Faculty of Medical Sciences, The University of the West Indies, St. Augustine 685509, Trinidad and Tobago; trudee.hoyte@sta.uwi.edu (T.H.); nikao.davis@my.uwi.edu (N.D.); candice.deonarinesingh@my.uwi.edu (C.D.); amir.desilva@my.uwi.edu (A.D.S.); dylan.dhanpaul@my.uwi.edu (D.D.); chloe.dookhoo@my.uwi.edu (C.D.); justin.doorpat@my.uwi.edu (J.D.); alexei.dopson@my.uwi.edu (A.D.); joash.durgapersad@my.uwi.edu (J.D.); arlene.williams@sta.uwi.edu (A.F.-A.W.-P.); rodolfo.fundora@sta.uwi.edu (R.A.-F.); 2Tulane National Primate Research Centre, Tulane University, Covington, LA 70433, USA; cpalmer3@tulane.edu; 3Laval, QC H7E 2Z8, Canada; odalis_asin@yahoo.es

**Keywords:** clinical diagnosis, immunodeficiency, systematic review, immunoglobulins

## Abstract

Transient hypogammaglobulinemia of infancy (THI) is a primary immunodeficiency caused by a temporary decline in serum immunoglobulin G (IgG) levels greater than two standard deviations below the mean age-specific reference values in infants between 5 and 24 months of age. Preterm infants are particularly susceptible to THI, as IgG is only transferred across the placenta from mother to infant during the third trimester of pregnancy. This study aimed to conduct a systematic review of the diagnostic criteria for transient hypogammaglobulinemia of infancy. Systematic review: Three electronic databases (PubMed, MEDLINE, and Google Scholar) were manually searched from September 2021 to April 2022. Abstracts were screened to assess their fit to the inclusion criteria. Data were extracted from the selected studies using an adapted extraction tool (Cochrane). The studies were then assessed for bias using an assessment tool adapted from Cochrane. Of the 215 identified articles, 16 were eligible for examining the diagnostic criteria of THI. These studies were also assessed for bias in the six domains. A total of five studies (31%) had a low risk of bias, while four studies (25%) had a high risk of bias, and bias in the case of seven studies (44%) was unclear. We conclude that THI is only definitively diagnosed after abnormal IgG levels normalise. Hence, THI is not a benign condition, and monitoring for subsequent recurrent infections must be conducted. The diagnostic criteria should also include vaccine and isohaemagglutinin responses to differentiate THI from other immunological disorders in infants.

## 1. Introduction

Transient hypogammaglobulinemia of infancy (THI) is a primary immunodeficiency caused by a temporary decline in the serum levels of immunoglobulin G (IgG), affecting infants between 5 and 24 months of age [1]. The foetus receives maternal antibodies in the womb; however, it starts producing its own antibodies from the 6th month of pregnancy until birth, which allows for a continual supply of IgG. In some cases, infants may have low IgG levels due to various factors, which makes them vulnerable to recurrent infections. Severe life-threatening infections are atypical because THI is a self-correcting immunodeficiency when serum IgG, IgA, and IgM concentrations normalise. However, severe symptoms include severe pneumonia, opportunistic infections caused by fungi or Staphylococcus, gastrointestinal problems, and bloodstream infections [2]. Most infants with THI do not have the standard findings that are found in other immunodeficiencies. Hence, infants are diagnosed with THI when their IgG levels are below two standard deviations from the mean for their age [3]. While the incidence level of THI is unknown, scientists worldwide have described it as being considerably underdiagnosed. This is because immunoglobulin levels are not usually measured in normal infants [4]. As such, THI is mainly diagnosed retrospectively, and the actual incidence of asymptomatic THI may be considerably higher than that recorded. THI have been reported to have a male-to-female ratio of 2:1. Approximately 60% of diagnoses are conducted by the age of 1 year, and the remaining 40% afterwards—often, not until 5 or 6 years of age [5]. The purpose of this study was to conduct a systematic review of THI diagnoses.

This study is important with respect to public health research because there is currently a lack of studies on THI in infants from the Caribbean, particularly in Trinidad and Tobago. Therefore, we believe that it is important to conduct this systematic review to critically appraise the limited existing literature to help future researchers and clinicians build a foundation for more robust research in the region. Our study is relevant to public health, as it is our hope that this review will educate primary care stakeholders regarding the criteria for diagnosing THI and will aid in earlier diagnoses, such that any life-threatening complications that typically arise from this disorder will be avoided or reduced, which will allow infants with THI to live a more satisfactory and adequate life while their IgG levels normalise.

Transient hypogammaglobulinemia of infancy (THI) is defined by the World Health Organization (WHO) and the International Union of Immunological Societies as a primary heterozygous immunodeficiency disorder characterised by reductions in both immunoglobulin G (IgG) and immunoglobulin A (IgA). Ameratunga et al. (2019) reported that it is difficult to determine IgA levels in infants. Therefore, IgG levels are often used as the main indicator of THI [3]. It has been reported that THI is resolved in children before the age of four [3]. Preterm infants are particularly susceptible to THI, as IgG is only transferred across the placenta from the mother during the third trimester of pregnancy. THI marks the occurrence of the lowest physiological levels of IgG as a result of reductions in maternal IgG via the placenta and the synthesis of IgG by the infant [3].

Additionally, in a clinical study conducted in 2008, THI accounted for 18.5% of all primary immunological disorders in Japan [4]. THI can be diagnosed if there is a decrease in serum IgG levels greater than two standard deviations below the mean age-specific reference values [6]. It should be stressed that a true diagnosis of THI can only be made if IgG levels have returned to normal [3]. THI is manifested by recurrent infections, low IgG, and sometimes low IgA, as well. There are several PIDs that share features with THI, including CVID, mild cases of XLA, IgG subclass deficiencies, and IgA selective deficiency. However, patients aged 4–6 months may present with clinical characteristics of THI. After 1 or 2 years these individuals may stop having recurrent infections, allergies, diarrhoea, and other signs and symptoms of THI. Importantly, these patients may have normal quantities of IgG and IgA. This would suggest that the disease has been cured. In this scenario, one can retrospectively make the diagnosis of THI, since Ig levels have returned to normal and the patient is clinically well [3]. Other PIDs will continue clinically manifesting in the patient, e.g., CVID and XLA, and those in which immunopathogenesis is not related to a transitory deficit of immunoglobulin according to the definition of THI by the IUIS.

Immunological tests and studies were carried out for each patient, including responses to vaccines and isohaemagglutinin blood tests [3]. Isohaemagglutinin blood tests are performed as differential diagnostic criteria for common variable immunodeficiency (CVID) and are also used to distinguish CVID from THI [3]. The absence of isohaemagglutinins is common in CVID, whereas it is normal in THI [3]. Based on these comparisons, it was noted that X-linked agammaglobulinemia is another differential diagnosis of THI, characterised by mutations in Bruton’s tyrosine kinase (BTK), as indicated in genetic studies. THI and Bruton’s agammaglobulinemia are comparable because both are prominent in males and share similar symptoms, such as recurrent bacterial infections [1]. The vaccine responses measured were specific to diphtheria-tetanus toxoid, *Haemophilus influenzae* type B (HIB), and pneumococcal polysaccharide [3].

Wang et al. (2004) outlined various complications associated with THI, such as recurrent sinopulmonary infection, prolonged fever, failure to thrive, atopic dermatitis, allergic rhinitis, asthma, and chronic diarrhoea [7]. Vaillant and Wilson (2021) also listed complications such as fungal infections, including oral candidiasis, bacteraemia, premature death, and neutropenia [1].

Antibiotic prophylaxis was effectively used in THI patients in one study, of which only four out of twenty-five (16%) children responded poorly to antibiotics. Allergen-specific immunotherapy was performed on three of the patients in late childhood to reduce their upper respiratory infections and allergies [3]. Duse et al. (2010) proposed an alternative form of treatment for THI in children: intravenous immunoglobulin therapy (IVIG) [6]. IVIG was then given every three weeks for a two-to-three-month period, with follow-up for three years. The results indicated that the treatment was highly effective, as the infections decreased tenfold and no severe infections were reported [6].

The prevalence of THI in the Caribbean region remains unknown. This may be due to the lack of clinical immunologists in this region. This study serves as a foundation for a deeper study on THI prevalence in the region. We are not aware of the existence of a previous study on the diagnosis of THI in the Caribbean, including in the Republic of Trinidad and Tobago. However, these patients were treated by family physicians or paediatricians. They used antibiotics and gammaglobulins to treat recurrent infections. They referred to these children or adolescents as “immunocompromised”.

### 1.1. Research Question

From critically appraising the existing literature, what is/are the clinical assessment(s) used to make a diagnosis of THI among infants between the ages of 4 and 36 months?

### 1.2. Research Aims and Objectives

This research focused on critically appraising the existing literature related to the diagnosis of transient hypogammaglobulinemia in infants.

The research objectives of this study were as follows:To systematically review the diagnostic criteria for THI.To identify gaps in the existing literature related to the diagnostic criteria for THI.

## 2. Methodology

### 2.1. Eligibility Criteria

Eligibility criteria is shown as Table 1.

### 2.2. Exclusion Criteria

Infants diagnosed after 36 months (three years);Studies that did not use a screening tool/criterion to diagnose;Studies that were abstracts only (i.e., did not include the full text).

### 2.3. Information Sources

The databases used in this study were PubMed, Medline, and Google Scholar. They were searched from September 2021 to April 2022.

### 2.4. Search Strategy

Databases were searched using keywords and terms relevant to the purpose of the study:Screening tool OR diagnosis AND transient hypogammaglobulinemia AND infants;Transient hypogammaglobulinemia of infancy AND diagnosis.

Filters were also applied using the search terms listed above for the PubMed and MEDLINE databases. These included free full-text availability, humans, English, and no restrictions on sex, publication time, or article type. Google Scholar database was searched using the terms “Transient Hypogammaglobulinemia of infancy” AND diagnosis.

### 2.5. Selection Process

The selection process involved the manual screening of titles and abstracts of studies and articles that met the inclusion criteria. This process involved searching the three chosen databases for articles by all eight reviewers using the search strategy detailed above (see Figure 1).

### 2.6. Data Collection Process

Each report/study was reviewed by two reviewers who worked independently of each other to avoid bias and to determine whether the articles met the inclusion criteria. A third reviewer was appointed to manage disagreements between the first two reviewers, if required. This process was conducted according to the PRISMA (Preferred Reporting Items for Systematic Reviews and Meta-Analyses) flowchart (Figure 1). Once the articles had been screened and were deemed to fit the inclusion criteria, a data extraction tool was used to collect information from the selected articles. The data collection tool that was used in this study was developed by the Cochrane Developmental, Psychosocial and Learning Problems Review Group [8]. This tool was adapted to fit our study using applicable sections. Furthermore, the information extracted from the studies included general information, study characteristics, participant characteristics, screening tools/diagnostic assessments/outcomes collected, statistical analysis, results, and conclusions.

### 2.7. Study Risk of Bias Assessment

Risk of bias was assessed using an assessment tool developed by the Cochrane Developmental, Psychosocial, and Learning Problems Review Group [8]. This tool assesses selection, performance, detection, attrition, reporting, and other types of bias. The risk of bias was assessed by two reviewers, acting independently of each other. A third reviewer was consulted only if there was disagreement between the first two reviewers. This assessment was then inputted into the online software Robvis v0.3.0 [9], which then used the “Generic Template” to generate high-quality figures that summarised the bias assessments of the eligible articles.

### 2.8. Ethical Considerations

This systematic review received ethical approval from the University of the West Indies, St. Augustine Campus, Ethical Committee on 11 March 2022, with an amendment made and approved by the committee on 17 May 2022 (CREC-SA.1491/04/2022).

## 3. Results

### 3.1. Study Characteristics

Of the eligible studies, ten (63%) were retrospective, four (25%) were prospective, one (6%) was a literature review, and one (6%) was a report. These studies were conducted in countries such as New Zealand [3], Turkey [10,11,12,13], Taiwan [7], Italy [4,14,15,16], Poland [17], Iran [18], China [19], the US [20], France [21], and the United Kingdom [22] during the period of 1995–2021. All 16 reports included immunological studies involving the testing of IgG levels, which plays a dominant role in the diagnostic criteria for THI. Two studies [17,18] cited formal reference criteria used to diagnose immunodeficiencies: the European Society for Immunodeficiency (ESID) diagnostic criteria [23], the Pan-American Group for Immunodeficiency (PAGID) [24], and the International Union of Immunological Society [25]. The remaining 14 studies, however, reported various clinical assessments that were conducted to diagnose THI but were not an established set of clinical guidelines. Of the reviewed studies, as shown in Table 2, a diagnosis of THI was made considering the age of the patients (5 months to 3 years). Confirmatory diagnosis was performed after a laboratory report confirmed normal IgG levels [1,3,4,11,12,15,17,19,21].

### 3.2. Risk of Bias in Studies

All sixteen eligible studies were assessed for bias in the six domains, which included selection, performance, attrition, reporting, detection, and other biases. Of the sixteen studies, five (31%) had a low risk of bias, four (25%) had a high risk of bias, and bias was unclear in seven (44%) cases (see Figure 2). The studies and their assessed domains of bias were as follows: selection bias [3,4,7,11,12,13,14,17,18,19,21], performance bias [7,11,13], attrition bias [3,11,12,17], reporting bias [4,17,21], detection bias [4], and other types of bias, such as precision bias [4,11,12].

## 4. Discussion

Among the 16 analysed studies, a plethora of criteria were used to diagnose THI in infants, with the most common being abnormally low levels of IgG. Duse et al. (2010) mentioned that THI can be diagnosed if there is a decrease in serum IgG levels greater than two standard deviations below the mean age-specific reference values [6]. This concept was adhered to in 12 of the 16 studies [1,3,4,7,11,12,13,14,16,17,18,21]. As a result, a decrease in serum IgG proved to be the most important diagnostic marker for THI. It is also the most consistent and well-known criterion for the identification of a wide variety of immunological disorders, such as common variable immunodeficiency (CVID) and X-linked agammaglobulinemia (XLA), which also present with decreases in serum IgG immunoglobulins [20]. However, considering the appraised studies, a lowered serum IgG level was the most universally accepted criterion for diagnosing THI. None of the published case reports and series strictly adhered to the IUIS definition of THI as having decreased levels of both IgG and IgA [3], and this criterion can be used for an initial or suspected diagnosis. To definitively diagnose THI, Ameratunga et al. (2019) suggested that THI can only be diagnosed retrospectively after IgG levels have normalised, since many adolescents and adults are cured of THI beyond infancy after having multiple hospitalisations, due to severe infections in some cases [3]. Four other studies also indicated this aspect of THI [4,12,17,19]. This adds to the growing body of evidence that THI can only be definitively identified when IgG levels are normalised, including in infants. In several of the analysed studies [3,4,11,19], it was suggested that THI does not persist after the age of four, thereby indicating that THI is a transitory PID disorder characterised by mild to severe recurrent infections of the respiratory system and sinus, atopy, and diarrhoea. In a few cases, failure to thrive resolves during childhood. The results also suggest that levels of other immunoglobulins (IgA and IgM) can be used as parameters, along with IgG, in the detection of THI [14,15,19,22], as in the definition of THI by IUIS, which again stated that it is a disorder of both low IgG and IgA in a child [3]. Even studies that used IgA and IgM levels as diagnostic criteria still included individuals with both low and normal IgA and IgM levels, in addition to using IgG levels as the main indicator [11,16]. Therefore, the validity of these results remains. However, this reflects the lack of IUIS policies [3] in terms of the definition of THI. We agree with the discrepancies in the literature because we appraised the literature on THI from the period 2000–2022 (22 years) for new policies on PID by the IUIS and laboratory developments.

“According to the proposal by the European Society for Immunodeficiencies and the Pan American Group for Immunodeficiency in 1999, CVID was defined as follows: CVID is probable in a male or female patient who has a marked decrease in IgG (at least 2 SD below the mean for age) and a marked decrease in at least one of the isotypes IgM or IgA, and fulfils all of the following criteria:Onset of immunodeficiency at greater than 2 years of ageAbsent isohaemagglutinins and/or poor response to vaccinesDefined causes of hypogammaglobulinemia were excluded according to a list of differential diagnosis” [26].

Kornfeld et al. (1995) published a curious case of a child with an initial diagnosis of THI and CVID hiding X-linked agammaglobulinemia or Bruton’s disease. We considered this case study important, even when only one patient was evaluated, to bring the awareness of the scientific community to the many faces that THI might have, and the importance of a good immune system assessment and differential diagnosis.

THI may cause recurrent infections by encapsulated bacteria such as Staphylococcus aureus and Streptococcus. These infections can be treated with antibiotics, such as Amoxicillin, as follows: children, 10–20 mg/kg per day, single dose, or divided into 2× (maximum: 875 mg/day); adolescents and adults: 875 mg. Amoxicillin and clavulanate: Children were administered a single dose of 20 mg/kg per day or divided into 2× (maximum: 875 mg/day based on amoxicillin) [27].

Moreover, several studies included diagnostic criteria that were not found in other identified studies [4,21], such as T-cell defects and in vitro immunoglobulin production. T cell defects were used as exclusion criteria to differentiate between THI and other predominant antibody deficiencies. The measurement of in vitro immunoglobulin production was based on the ability of B cells to produce IgG and could serve in the differential diagnosis and as a prognostic value.

It is said that THI is resolved in children before the age of four, but THI is an antibody deficiency. Taking this into consideration it is reasonable to speculate that some B-cell developmental processes are adversely affected, causing disruption in normal antibody production, which is expected to occur fully by the age of one year. During gestation, the mother provides the highest quantities of antibodies (IgG) that cross the placentae to temporarily reconstitute the humoral immune system of the neonate. B-cells are transformed into plasma cells, which produce IgG as well as IgM, IgA, IgE, and IgD. T-cells play an essential role in the antibody switching from IgM to other types of Igs. In THI, there may be deficiencies between T- and B-cell cooperation. For example, T-helper cells 1 (Th-1) may not synthesize enough IL-2, or Th-2 cells may not produce enough IL-4, IL-5 and IL-6, which play important roles in B cell proliferation and/or differentiation, and therefore the synthesis of several immunoglobulins is affected [28]. In addition, the malfunctioning of T- and B-cell receptors can cause a poor recognition of antigens, including thymus-dependent or thymus-independent antigens. These processes are gradually developed in neonates until they can maturate immunologically and gradually produce their own antibodies [1,3,7,10]. The time frame to achieve immune system maturity in most individuals is from 1 to 4 years, but this may require more time, and so THI can manifest clinically as recurrent infections by extracellular microorganisms such as bacteria, and low antibody production post exposure to immunogens and vaccines. However, a good differential diagnosis should be made, because XLA and CVID could mimic THI [1,20,26].

Several factors for the late recovery of Igs in THI patients have been proposed and include (1) the occurrence of recurrent infections (>6/year), (2) shorter breastfeeding duration and (3) initial low IgA and IgM levels [29].

Data on the incidence and prevalence of THI are scarce despite many studies on THI over the years. Table 3 summarises the incidence of THI in selected countries.

“Since that first description of THI there have been many reports of both single cases and cohorts of infants with low levels of Igs, and THI has been included in the classification of the Primary Immunodeficiencies provided by the International Union of Immunological Societies (IUIS). In early reports, THI was a rare condition, with several papers noting scant numbers of cases compared with other immune deficiencies. Later, however, probably reflecting greater awareness of the diagnosis, a larger series of cases began to be reported, and a greater relative frequency of THI” [33].

A differential diagnosis of THI must be carried out, as it can only be definitively diagnosed retrospectively. Eight studies included vaccine responses based on the diagnostic criteria [1,3,4,11,12,13,20,22]. The quality of these vaccine responses was also measured and concerned diphtheria, tetanus toxoid, *Haemophilus influenzae* type B (HIB), and pneumococcal polysaccharide. In the case of THI, there is usually a normal protective response to these vaccines. However, the key factor of these differential diagnoses, such as CVID [26] and XLA, is that they exhibit no vaccine responses [3,22,34,35,36]. Vaccine responses represent a distinct diagnostic criterion for differentiating THI, XLA, and CVID. However, vaccine responses are only beneficial as diagnostic criteria when evaluated before the occurrence of IgG normalisation in THI.

Diphtheria, tetanus toxoid, and *Haemophilus influenzae* type B (HIB) vaccines are not accurate for measuring the immune response in some patients with PID [3]. The ability to mount an IgA- or IgG-mediated response against pneumococcal polysaccharide antigens, or the ability to maintain the antibody response over time, identify a normal protective response to this vaccine [37,38,39]. Patients with CVID are IgA-poor responders to pneumococcal polysaccharide antigens [38].

Furthermore, four studies identified the isohaemagglutinin response as a diagnostic criterion to differentiate between CVID and THI [1,3,11,12]. It has also been suggested that isohaemagglutinin antibody production is a more reliable criterion than vaccine response [3]. Its status is considered normal, and its detection is based on the production of anti-A and anti-B antibodies. The absence of isohaemagglutinins is common in CVID, whereas they are normally produced in THI [3]. Similarly, for vaccine responses, this criterion serves a useful role in distinguishing THI before IgG normalisation. In addition, the use of lymphocyte subsets as a diagnostic criterion has been discussed in nine of the analysed studies [4,7,11,13,14,19,20,21,22]. Particular attention was given to the B-cell compartments, as several cohort studies have compared their values to those of healthy individuals. B-cell subsets can be used to evaluate THI prognosis as well as to differentiate it from XLA. A laboratory investigation of a patient with transient hypogammaglobulinemia of infancy includes the assessment of immunoglobulin levels and B-cell number and functions. B-cell subset studies include the assessment of CD19 and CD20 markers using flow cytometry and B-cell antibody responses to antigens [1]. The same studies can also be applied to patients with XLA.

T-cell defects are a group of disorders characterised by malfunction or low numbers of T cells. THI may be cause by an underlying T-cell defect [28,40].

THI is not associated with ethnicity and appears to affect people of all regions of the world [34]. Data on the worldwide incidence of THI is lacking, which underscores the need for further research in this field.

The inconsistencies in published THI reports include the following:
Prematurity (very low IgG levels at birth);Inclusion of hypogammaglobulinemic children beyond 2 years of age;Selection of neonates or infants with a family history of immunodeficiency;Inconsistencies in the immunoglobulin evaluation;Inconsistencies in the selection of cases and their differential diagnosis.

The cumulative evidence showing that patients with THI exhibit clinical manifestations such as recurrent infections, atopy, and diarrhoea, which may require hospitalisation in some cases [1,3,7,14,18,19,21,41,42], suggest that THI is not a benign condition.

## 5. Conclusions

After critically appraising 16 studies, the results imply that THI may persist after the age of four and is mainly diagnosed retrospectively after the normalisation of IgG levels. Likewise, the findings of these studies consistently show decreases in serum IgG levels greater than two standard deviations below the mean age-specific reference as an initial diagnosis of THI. Vaccines, isohaemagglutinin responses, and B- and T-cell subsets are thus additional clinical evaluations that are conducted to differentiate XLA, CVID, and THI prior to the normalisation of IgG. THI is not a benign condition and must be monitored closely; if symptomatic, appropriate treatment with antibiotics and immunoglobulin replacement therapy should be institutionalised.

The World Health Organization/International Union of Immunological Societies (WHO/IUIS) definition mandates a reduction in both IgG and IgA to qualify for THI [36].

## Figures and Tables

**Figure 1 children-10-01358-f001:**
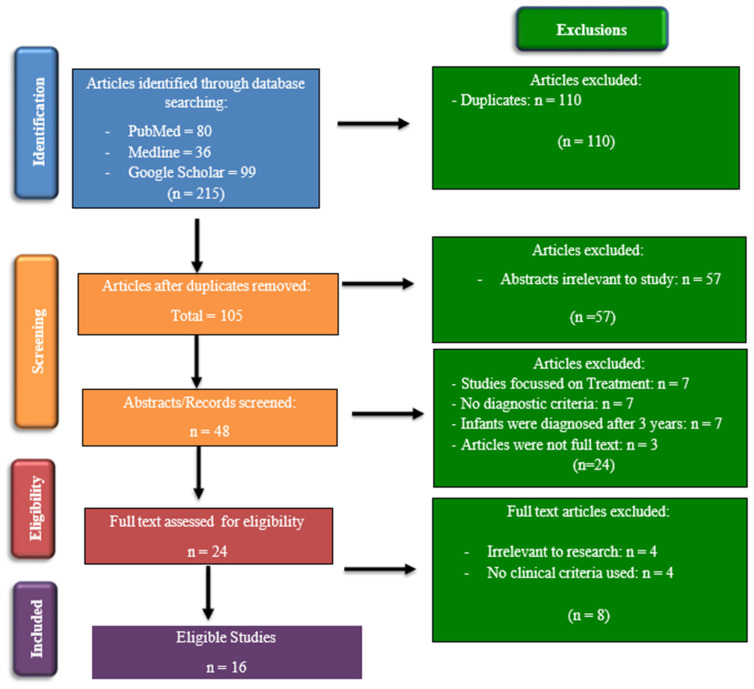
PRISMA flow chart showing the process for identification and inclusion of the studies.

**Figure 2 children-10-01358-f002:**
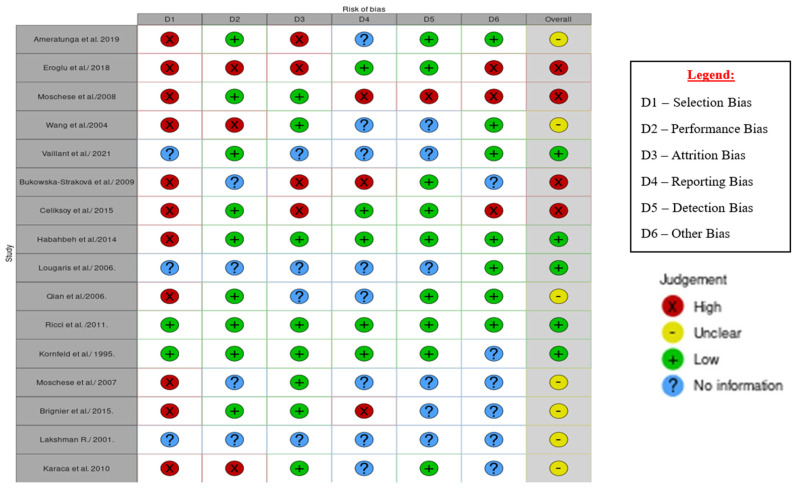
Traffic light plot of the results of bias assessment of the 16 individual studies [1,3,4,7,11,12,13,14,15,16,17,18,19,20,21,22].

**Table 1 children-10-01358-t001:** The eligibility/inclusion criteria for study, using the PICOS format.

Population	Intervention	Comparison	Outcome	Study Type
Infants (human children aged 4–36 months)	Screening tool or diagnostic criteria used to make a diagnosis	N/A	Clinical diagnosis (the criteria or clinical assessment used to diagnose THI)	All study types

**Table 2 children-10-01358-t002:** An overview of the 16 eligible studies examining the diagnosis of THI.

Author/Year	Aim	Type of Study/Design	Participants	Diagnostic Criteria Used/Referenced
Ameratunga et al., 2019 [3]	To determine the clinical features and recovery for patients with THI	Retrospective case series	47 patients < 4 years (at the time of diagnosis)	History, examination, immunological studies, vaccine response, and isohaemagglutinin production
Eroglu et al., 2018 [11]	To analyse the B-cell subsets of patients with a THI diagnosis and compare with healthy age-matched Turkish children	Retrospective cohort study	20 patients with comparison between THI and healthy aged- matched children	Low levels of IgG (<2 SD) with/without decreases in IgA or IgM. Lymphocyte subsets, isohaemagglutinins, and vaccine responses. The exclusion of the defined causes of secondary hypogammaglobulinemia
Moschese et al., 2008 [4]	To characterise the clinical and immunological features of children with THI and to assess the predictive parameters of clinical evolution	Prospective cohort study	77 THI children at initial diagnosis and 57 patients at follow-up	Exclusion of other causes of hypogammaglobulinemia. Diagnosis conducted with the normalisation of IgG levels. Memory B-cell subsets and in vitro immunoglobulin production was evaluated
Wang et al., 2004 [7]	To review clinical features and the outcome of children with primary hypogammaglobulinemia	Retrospective, case–control study	33 patients	Quantifications of serum immunoglobulins (IgA/IgM/IgG) and lymphocyte subsets were performed
Vaillant and Wilson et al., 2021 [1]	To review the clinical presentation, epidemiology, pathophysiology, and treatment of THI	Literature review	Clinical book chapter about all aspects of THI	Low serum IgG as well as the detection of isohaemagglutinins (IgM) and IgG antibodies (post-exposure)
Bukowska-Straková et al., 2009 [17]	To evaluate the B-cell compartment in the peripheral blood of children with different types of hypogammaglobulinemia	Retrospective, longitudinal, observational study	600 children with immunodeficiencies. Additionally included 28 adults with CVID and 12 healthy controls	Used the criteria of the International Union of Immunological Societies in addition to patients whose level of Ig were normalised before age 4 for THI
Celiksoy et al., 2015 [12]	To analyse the memory of the B-cell subsets of patients with antibody deficiencies	Retrospective study	67 patients (20 patients with THI aged 1–3). A total of 28 healthy children of matching ages were also included	Low serum IgG levels; low IgA and/or IgM levels upon admission; normalisation of low Ig levels during follow-up; the normal production of an antibody specific to isohaemagglutinins; and an intact cellular immunity
Lougaris et al., 2006 [15]	Not stated	Report	Reviewed the comparison of the Ig value with age- matched controls	Lab analysis of the serum IgG, IgM, and IgA. Differential diagnosis between these two conditions (THI and CVID) could not be made with certainty before 2–3 years of age
Qian et al., 2006 [19]	To determine the clinical signs, immunological changes, and the outcomes with hypogammaglobulinemia	Prospective	91 patients < 2 years with warning signs of PID	Serum immunoglobulin and lymphocyte subsets were analysed. The normalisation of Ig levels in follow-up visits was noted in order to confirm the diagnosis
Ricci et al., 2011 [16]	To assess the clinical and immunological evolution of premature and full-term infants with hypogammaglobulinemia	Prospective, cross-sectional study	24 children (11 premature and 13 full-term infants)	Reduction in IgG with/without a reduction in IgA and IgM
Kornfeld et al., 1995 [20]	To compare and diagnose a patient with XLA that presented with an initial diagnosis of THI and CVID	Retrospective case report	1 patient	Low Ig levels, the vaccine response (to both diphtheria and tetanus), and the circulating B cells were evaluated
Moschese et al., 2007 [14]	To determine if memory B-cell subsets can be used as a predictive marker for the THI outcome	Retrospective cohort study	36 patients with comparison of THI and healthy patients	Serum IgG levels < 2 SDs, circulating B cells > 2%, and an exclusion of known causes of secondary hypogammaglobulinemia
Brignier et al., 2015 [21]	To define hypogammaglobulinemia	Retrospective cohort study	44 patients with early onset (i.e., <6 yrs.)	Serum immunoglobulin levels and T-cell defects were characterized. Exclusion of other PADs
Lakshman R. 2001 [22]	To discuss the clinical presentation, laboratory diagnosis, and management of hypogammaglobulinemia	Cross-sectional study	Management of hypogammaglobulinemia	A full blood count and peripheral smear examination, as well as quantitative estimations of serum Ig (IgG, IgA, IgM, and IgE), IgG subclass estimation, lymphocyte subset estimation, and the responses to vaccines (both diphtheria and tetanus)
Karaca et al., 2010 [13]	To evaluate the clinical and immunological data and outcomes	Retrospective, cross-sectional study	101 patients	Measurement of the serum immunoglobulins (IgG, IgA, and IgM) as well as recording of vaccine responses and lymphocyte subpopulations
Habahbeh et al., 2014 [18]	To describe the clinical spectrum of primary antibody deficiency in order to increase awareness for early referral	Retrospective study using medical records	Medical records of 53 paediatric patients, 19% of which possessed THI	Pan-American Group for Immunodeficiency (PAGID) and the European Society for Immunodeficiency (ESID) diagnostic criteria. Molecular diagnosis not available at the hospital used in this study

**Table 3 children-10-01358-t003:** Incidence/prevalence of THI in selected countries.

Incidence of THI/Country	References
23 per 10^6^ births/Australia	[30]
0.061–1.1 per 1000 live births/Japan	[2]
11 per 10^4^ births/USA	[31]
**Prevalence of THI/Country**	
9.8%/Northern India	[32]
10% of all cases of hypogammaglobulinemia/Taiwan	[33]
18.5% of all cases of PID (the most prevalent/Japan	[33]

The estimated frequency of transient hypogammaglobulinemia varies among studies. In some studies, IgG deficiency was most common in childhood. Research has described it worldwide, and it is thought to be highly underdiagnosed owing to variations in the diagnostic criteria [1,3].

## Data Availability

Not applicable.

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
