# Peer review of "A Systematic Review of the Clinical Diagnosis of Transient Hypogammaglobulinemia of Infancy"

_children, 2023, doi:10.3390/children10081358_

Round 1
Reviewer 1 Report
Could the authors be more specific regarding the incidence and prevalence of hypogammaglobulinemia in Carribean and Trinidad Tobago region?
It would be of great interest to exactly state the type of lab exams performed to confirm the existence or not of a CVID not only those related to transient hypogammoglobulinemia.
In the cited paper there is one that comprises of 1 pacient Kornfeld et al. compared to all the other studies. Could the author state why did they consider the study important?
What were the most used antibiotics in the study other than immunoglobulins at these particular group of children with THI? What was the dose regimen of those antibiotics?
Author Response
Could the authors be more specific regarding the incidence and prevalence of hypogammaglobulinemia in Caribbean and Trinidad Tobago region?
The prevalence of THI in the Caribbean region is unknown. It might be cause due to the lack of clinical immunologists in the region. This study serves as a foundation for a deeper study on THI prevalence in the region. We are not aware of the existence of a previous study on the diagnosis of THI in the Caribbean including The Republic of Trinidad and Tobago. However, these cases are treated by family physicians or pediatricians. They use antibiotics to treat the recurrent infections, multivitamins and gammaglobulins. They call these children or adolescence as “immunocompromised”.
It would be of great interest to exactly state the type of lab exams performed to confirm the existence or not of a CVID not only those related to transient hypogammaglobulinemia.
“According to the proposal by the European Society for Immunodeficiencies and the Pan American Group for Immunodeficiency in 1999, CVID was defined as follows: CVID is probable in a male or female patient who has a marked decrease of IgG (at least 2 SD below the mean for age) and a marked decrease in at least one of the isotypes IgM or IgA, and fulfills all of the following criteria:
- Onset of immunodeficiency at greater than 2 years of age
- Absent isohemagglutinins and/or poor response to vaccines
- Defined causes of hypogammaglobulinemia have been excluded according to a list of differential diagnosis” [26].
In the cited paper there is one that comprises of 1 patient Kornfeld et al. compared to all the other studies. Could the author state why did they consider the study important?
Kornfeld et al. 1995 published a curious case of a child with the initial diagnosis of THI and CVID hiding an X-linked agammaglobulinemia or Bruton’s disease. We considered this case study important, even when only one patient was evaluated, to make aware the scientific community of the many faces that THI migh thave, and the important of a good immune system assessment, and differential diagnosis to see the light at the end of the tunnel. I mean a wrong diagnosis was corrected after a good scientific judgment [20].
What were the most used antibiotics in the study other than immunoglobulins at these particular group of children with THI? What was the dose regimen of those antibiotics?
Amoxicillin: Children: 10–20 mg/kg per day, single dose or divided into 2× (maximum: 875 mg/day). Adolescents and adults: 875 mg. Amoxicillin and clavulanate: Children: 20 mg/kg per day single dose or divided into 2× (maximum: 875 mg/day based on amoxicillin).
Reviewer 2 Report
References - the are only the websites of some articles. Are they a scientific article?
line 73 "...in the lat and third trimester of pregnancy." - one of this description is sufficient.
line 100: "...four of the children..." - four of how many? maybe percent will be better?
Author Response
References - the are only the websites of some articles. Are they a scientific article?
There are 19 scientific articles (76%). Other references are relevant to websites that cover immunodeficiency disorders including transient hypogammaglobulinemia of infancy (THI) or methodological aspects of a systemic review.
line 73 "...in the last and third trimester of pregnancy." - one of this description is sufficient.
We wrote “ in the third trimester of pregnancy”.
line 100: "...four of the children..." - four of how many? maybe percent will be better?
4 out of 25 (16%) children responded poorly to the antibiotics.
Reviewer 3 Report
Justiz-Vaillant A et al present a review of the literature around the diagnosis of THI, a benign medical condition that needs to be differentiated from other immunological disorders. The topic of the manuscript is of interest and reflects the agreement gap at present. Authors have done a good job at reviewing the literature and providing a summary of it for the reader. However, the text as written is not sufficiently interpretive and is not clear what point they are trying to make. Maybe partial conclusions that summarize main ideas extracted from the interpretation of the literature provided would help.
In addition, there are several points that need revision before acceptance:
-
The objective of the study is not stated clearly on the abstract
-
Line 71: “It is said that THI is resolved in children before the age of four”. Please, explain possible reasons that could be involved in this statement, such as immune system development.
-
Lines 76-77: “In a Turkish study, Mamedov et al. (2013) outlined the prevalence of THI to be 7.01 per 100,000 individuals [1]” Please clarify if this prevalence is in Turkey. There is an association of this disease with ethnicity? There is a world Wide incidence? Or are there countries where the incidence is higher?
-
lines 81-82: “It should be stressed that a true diagnosis of THI can only be made if the IgG levels have returned to normal [3].” Could you explain this statement a bit more please?
-
Lines 127-128: “was searched from September 2021–April 2022, while Medline and Google Scholar were searched from March–April 2022.” Why did you choose those periods of time? In addition, the periods of time stated on lines 137-138 and 178 are different.
-
Lines 185-187: “Of the reviewed studies (see Table 2), a diagnosis was made from about 5–6 months to 3 years in most cases, with a confirmatory diagnosis performed after the IgG levels had normalised [1,3,4,11,12,15,17,19,21]” Please, rephrase this paragraph since it is a bit confusing.
-
Table 2, pag 6 and 7. Bukowska-Straková et al. 2009 [17] and Lougaris et al. 2006 [15] Lakshman R. 2001 [22]: the number of children included on these studies is missing.
-
line 196: “Figure 3 shows a summary plot for the assessed bias in all 16 studies” I guess that the figure number is wrong, please check.
-
Lines 209-211; “However, considering the appraised studies, lowered serum of IgG was the most universally accepted criterion for diagnosing THI, although it was not the most accurate.” Why is it not accurate? Could you deepen this concept?
-
lines 228-240: “The differential diagnoses of THI must be ruled out, as it can only be definitively diagnosed retrospectively. From the results, two main responses were identified as factors that distinguish THI from other immunological disorders. Eight studies included vaccine responses among their diagnostic criteria [1,3,4,11–13,20,22]. The quality of these vaccine responses was also measured and concern diphtheria, tetanus toxoid, Haemophilus influenzae type B (HIB), and pneumococcal polysaccharide. In cases of THI, there is usually a normal protective response to these vaccines. The key differential diagnoses, however, such as CVID and XLA, exhibit differing responses. In one study, CVID was found to have a greater than level of protective response [3], whereas in another it was found to have a poor response [22]. Vaccine responses represent a distinct diagnostic criterion by which to differentiate between THI, XLA, and CVID. However, vaccine responses are only beneficial as a diagnostic criterion when evaluated before the occurrence of IgG normalisation”. You stated that the immune response against vaccines represents a distinct diagnostic criterion in order to differentiate among THI, XLA, and CVID. However, this concept is not clear in the preceding paragraph nor is it mentioned the characteristics of that response on XLA patients.
In addition, clarify how you defined “normal protective response to these vaccines”.
-
lines 257-258. “This criterion can be further used after the age of two to predict the prognosis of THI.” Did you find some evidence that supports this statement? Please, provide a citation.
-
Lines 254-259: “Most studies after the investigation stated that the B cell subset findings were not remarkably different between an infant with THI and a healthy infant [4,11,12,14,15]. Patients with persistent hypogammaglobulinemia after the age of four showed decreased levels of B cell subsets. This criterion can be further used after the age of two to predict the prognosis of THI. Therefore, B cell subsets can be excluded as a diagnostic criterion for THI. XLA can also be distinguished from THI as B cells are present in THI but absent in XLA.” This paragraph is confusing. Scientific evidence showed that B cell subsets can be used to evaluate THI prognosis as well as to differentiate it from XLA. However, you stated that B cell subsets can be excluded as a diagnostic criterion. Could you please explain your conclusions better?
-
lines 263-264: “T-cell defects were used as an exclusion criterion to differentiate between THI and other predominant antibody deficiencies” Could you please define T-cell defects?
-
lines 265-266: “However, they results vary, and this criterion cannot be considered to allow consistent conclusions.” What do you mean by “results vary”?
-
lines 282-286: “Hence, THI is mostly a benign condition, but monitoring must be conducted for subsequent recurrent infections, which may pose a severe risk to the health and quality of life of these infants. The diagnostic criterion should also include vaccine and isohaemagglutinin responses to allow differentiation from other immunological disorders in infancy.” This review in the actual form does not allow us to conclude that THI is "a benign condition".
English language needs revision
Author Response
Justiz-Vaillant A et al present a review of the literature around the diagnosis of THI, a benign medical condition that needs to be differentiated from other immunological disorders. The topic of the manuscript is of interest and reflects the agreement gap at present. Authors have done a good job at reviewing the literature and providing a summary of it for the reader. However, the text as written is not sufficiently interpretive and is not clear what point they are trying to make. Maybe partial conclusions that summarize main ideas extracted from the interpretation of the literature provided would help.
We appreciate the supportive comments by the reviewer on the merits of our review.
In addition, there are several points that need revision before acceptance:
- The objective of the study is not stated clearly on the abstract
We thank the reviewer for this important comment. We have included the following in line 20-22 (Page 1). “Thisstudy aimed at conducting a systemic review on the diagnostic criteria of the transient hypogammaglobulinemia of infancy”.
- Line 71: Please, explain possible reasons that could be involved in this statement, such as immune system development. Which statement is the reviewer referring to? You could put the statement here.
We thank the reviewer for this suggestion, which is addressed on lines 275 to 297, page 10.
“It is said that THI is resolved in children before the age of four” but THI is an antibody deficiency. Taking this into consideration it is reasonable to speculate that some B cell developmental processes are adversely affected causing disruption in normal antibody production, which is expected to occur fully by age one year. During birth, the mother provides the highest quantities of antibodies (IgG) that cross the placentae to temporarily reconstitute the humoral immune system of the neonate.
B cells are transformed into plasma cells, which produce IgG as well as IgM, IgA, IgE and IgD. T-cells pay an essential role in the antibody switching from IgM to other types of Igs. In THI there may be deficiencies between T- and B cells cooperation. For example, T-helper cells 1 (Th-1) may not synthesize enough IL-2, or Th-2 cells do not produce enough IL-4, IL-5 and IL-6, which play an important role in B cell proliferation and/or differentiation and therefore, the synthesis of several immunoglobulins is affected. In addition, malfunctioning of T- and B-cells receptors can cause poor recognition of antigens including thymus-dependent or thymus-independent antigens. These processes are gradually developed in neonates until they can maturate immunologically and gradually produce their own antibodies [1,3,7,10]. The time frame to achieve immune system maturity in most individuals is from 1 to 4 years but this may require more time, and so THI can manifest clinically as recurrent infections by extracellular microorganisms such as bacteria, and low antibody production post-exposure to immunogens and vaccines. However, a good differential diagnosis should be made, because XLA and CVID could mimic THI [1,20,26].
Several factors for late recovery of Igs in THI patients have been proposed and include (1) occurrence of recurrent infections (>6/year), (2) shorter breastfeeding duration and (3) Initial low IgA and IgM levels [28].
- Lines 76-77: “In a Turkish study, Mamedov et al. (2013) outlined the prevalence of THI to be 7.01 per 100,000 individuals [1]” Please clarify if this prevalence is in Turkey. There is an association of this disease with ethnicity? There is a worldwide incidence? Or are there countries where the incidence is higher?
We appreciate this important comment by the reviewer. To provide a wider view on THI incidence/prevalence we have now included data from selected countries in table 3.
Table 3. Incidence/prevalence of THI in selected countries
|
Incidence of THI/Country |
References |
|
23 per 106 births/Australia |
[30] |
|
0.061-1.1 per 1,000 live births/Japan |
[2] |
|
11 per 104 births/USA |
[31] |
|
Prevalence of THI/Country |
|
|
9.8 % /Northern India |
[32] |
|
10 % of all cases of hypogammaglobulinemia/Taiwan |
[33] |
|
18.5% of all cases of PID (the most prevalent/Japan |
[33] |
The estimated frequency of transient hypogammaglobulinemia varies between studies. In some studies, it is the most common IgG deficiency in childhood. Research has described it worldwide, and it is thought to be highly underdiagnosed due to variations in the criteria for diagnosis [1,3].
THI is not associated with ethnicity and appears to affect people of all regions of the world [34]. Data on the worldwide incidence of THI is lacking, which underscores the need for further research in this field. On page 12 lines 346-348.
- lines 81-82: “It should be stressed that a true diagnosis of THI can only be made if the IgG levels have returned to normal [3].” Could you explain this statement a bit more please?
- We appreciate the request for clarity which has been addressed on lines 81 and 91, page 2.
THI is manifested by recurrent infections, low IgG, and sometimes low IgA as well. There are several PID that share features with THI, including CVID, mild cases of XLA, IgG subclass deficiencies, and IgA selective deficiency. However, patients aged 4-6 months may present with clinical characteristics of THI. After 1 or 2 years these individuals may stops having recurrent infections, allergy, diarrheas, and other signs and symptoms of THI. Importantly, these patients may have normal quantities of IgG and IgA. This would suggest that the disease has been cured. In this scenario one can retrospectively make the diagnosis of THI, since Ig levels have returned to normal, and the patient is clinically well. Other PIDs will clinically continue manifesting in the patient e.g. CVID and XLA, and those which immunopathogenesis is not related to a transitory deficit of immunoglobulin according to the definition of THI by the IUIS.
- Lines 127-128: “was searched from September 2021–April 2022, while Medline and Google Scholar were searched from March–April 2022.” Why did you choose those periods of time? In addition, the periods of time stated on lines 137-138 and 178 are different.
We thank the reviewer for commenting about it. It was revised and addressed on line 141-142 and lines 151 and 152, respectively on page 4.
- Lines 185-187: “Of the reviewed studies (see Table 2), a diagnosis was made from about 5–6 months to 3 years in most cases, with a confirmatory diagnosis performed after the IgG levels had normalised [1,3,4,11,12,15,17,19,21]” Please, rephrase this paragraph since it is a bit confusing.
We appreciate the reviewer requesting clarity. We have now replaced this statement with the following lines 198-201 on page 6.
“Of the reviewed studies as shown in Table 2, a diagnosis of THI was made taking in consideration the age of the patients (5 months to 3 years). The confirmatory diagnosis was performed after a laboratory report assuring normal IgG levels [1,3,4,11,12,15,17,19,21]”
- Table 2, pag 6 and 7. Bukowska-Straková et al. 2009 [17] and Lougaris et al. 2006 [15] Lakshman R. 2001 [22]: the number of children included on these studies is missing.
We thank the reviewer for this important suggestion which is addressed in Table 2.
Bukowska-Straková et al. 2009 [17] included 600 children with PID.
Lougaris et al. 2006 [15] is a seminal review that highlights diagnostic criteria of THI
Lakshman R. 2001 [22] is a cross-sectional study about the management of Hypogammaglobulinemia.
- line 196: “Figure 3 shows a summary plot for the assessed bias in all 16 studies” I guess that the figure number is wrong, please check.
We thank the reviewer for spotting this oversight. This has now been corrected.
- Lines 209-211; “However, considering the appraised studies, lowered serum of IgG was the most universally accepted criterion for diagnosing THI, although it was not the most accurate.” Why is it not accurate? Could you deepen this concept?
We thank the reviewer for this suggestion. We have expanded this statement on lines 224-227 page 9.
“However, considering the appraised studies, lowered serum of IgG was the most universally accepted criterion for diagnosing THI. However, this does not strictly adhere to the IUIS definition of THI as having decreased levels of both IgG and IgA [33].
- lines 228-240: “The differential diagnoses of THI must be ruled out, as it can only be definitively diagnosed retrospectively. From the results, two main responses were identified as factors that distinguish THI from other immunological disorders. Eight studies included vaccine responses among their diagnostic criteria [1,3,4,11–13,20,22]. The quality of these vaccine responses was also measured and concern diphtheria, tetanus toxoid, Haemophilus influenzae type B (HIB), and pneumococcal polysaccharide. In cases of THI, there is usually a normal protective response to these vaccines. The key differential diagnoses, however, such as CVID and XLA, exhibit differing responses. In one study, CVID was found to have a greater than level of protective response [3], whereas in another it was found to have a poor response [22]. Vaccine responses represent a distinct diagnostic criterion by which to differentiate between THI, XLA, and CVID. However, vaccine responses are only beneficial as a diagnostic criterion when evaluated before the occurrence of IgG normalisation”. You stated that the immune response against vaccines represents a distinct diagnostic criterion to differentiate among THI, XLA, and CVID. However, this concept is not clear in the preceding paragraph nor is it mentioned the characteristics of that response on XLA patients.
We appreciate this suggestion and have made modifications accordingly.
“The differential diagnoses of THI must be ruled out, as it can only be definitively diagnosed retrospectively. Eight studies included vaccine responses among their diagnostic criteria [1,3,4,11–13,20,22]. The quality of these vaccine responses was also measured and concern diphtheria, tetanus toxoid, Haemophilus influenzae type B (HIB), and pneumococcal polysaccharide. In cases of THI, there is usually a normal protective response to these vaccines. The key differential diagnoses, however, for CVID and XLA is that they exhibit poor vaccine responses [3,22,35,36]. Vaccine responses represent a distinct diagnostic criterion by which to differentiate between THI, XLA, and CVID. However, vaccine responses are only beneficial as a diagnostic criterion when evaluated before the occurrence of IgG normalisation in THI”. On page 11 lines 312-321.
In addition, clarify how you defined “normal protective response to these vaccines”.
Diphtheria, tetanus toxoid, Haemophilus influenzae type B (HIB) vaccines were not all accurate to measure an immune response in some patients with PID [3]. The ability to mount an IgA- or IgG-mediated response against the pneumococcal polysaccharide antigens or the ability to maintain the antibody response over time identified a normal protective response to this vaccine [40-42]. Patients with CVID were IgA poor responders to pneumococcal polysaccharide antigens [41]. On page 11 lines 322-327.
- lines 257-258. “This criterion can be further used after the age of two to predict the prognosis of THI.” Did you find some evidence that supports this statement? Please, provide a citation.
Its statement is wrong. It is not supported by the literature.
" We appreciate this comment. There doesn't appear to be consensus with this statement in the field, so we have removed it"
- Lines 254-259: “Most studies after the investigation stated that the B cell subset findings were not remarkably different between an infant with THI and a healthy infant [4,11,12,14,15]. Patients with persistent hypogammaglobulinemia after the age of four showed decreased levels of B cell subsets. This criterion can be further used after the age of two to predict the prognosis of THI. Therefore, B cell subsets can be excluded as a diagnostic criterion for THI. XLA can also be distinguished from THI as B cells are present in THI but absent in XLA.” This paragraph is confusing. Scientific evidence showed that B cell subsets can be used to evaluate THI prognosis as well as to differentiate it from XLA. However, you stated that B cell subsets can be excluded as a diagnostic criterion. Could you please explain your conclusions better?
We agree with the reviewer. The paragraph has been deleted, and we will make clear that B cell subsets can be used to evaluate THI prognosis as well as to differentiate it from XLA. Laboratory investigation of a patient with transient hypogammaglobulinemia of infancy includes the assessment of immunoglobulin levels and B-cell number and functions. B cell subset studies include the assessment of CD19 and CD20 markers using flow cytometry and B cell antibody responses to antigens [1]. The same studies can also be applied to patients with XLA. On page 11 lines 337-343.
- lines 263-264: “T-cell defects were used as an exclusion criterion to differentiate between THI and other predominant antibody deficiencies” Could you please define T-cell defects?
We thank the reviewer for this important suggestion.
T-cell defects are a group of disorders characterised by malfunction or low numbers of T cells. THI may be cause by an underlying T-cell defect [28,40]. On page 11 lines 344-345.
- lines 265-266: “However, they results vary, and this criterion cannot be considered to allow consistent conclusions.” What do you mean by “results vary”?
We have deleted this statement to achieve clarity in the discussion.
- lines 282-286: “Hence, THI is mostly a benign condition, but monitoring must be conducted for subsequent recurrent infections, which may pose a severe risk to the health and quality of life of these infants. The diagnostic criterion should also include vaccine and isohaemagglutinin responses to allow differentiation from other immunological disorders in infancy.” This review in the actual form does not allow us to conclude that THI is "a benign condition".
We appreciate the reviewer’s concern here. According to various sources [1,3,18,14,19,21,40,41] the cumulative evidence showing that patients with THI exhibit clinical manifestations such recurrent infections, atopy, and diarrhoea, which may require hospitalisation in some cases suggest that THI is not a benign condition. On page 12 lines 355-357.

Round 2
Reviewer 1 Report
The authors have answered to all the questions that were addressed. Their answer has given a more profound insight regarding the region and also the treatment regimen that are given to infants and children with transient hypogammaglobulinemia of infancy.